

# Abrupt climate change as rate-dependent cascading tipping point

Johannes Lohmann[1], Daniele Castellana[2], Peter D. Ditlevsen[1], and Henk A. Dijkstra[2]

[1]Physics of Ice, Climate and Earth, Niels Bohr Institute, University of Copenhagen, Denmark
[2]Institute for Marine and Atmospheric Research, Utrecht University, The Netherlands

**Correspondence:** Johannes Lohmann (johannes.lohmann@nbi.ku.dk)

**Abstract.** We propose a conceptual model comprising a cascade of tipping points as a mechanism for past abrupt climate changes. In the model, changes in a control parameter, which could for instance be related to changes in the atmospheric circulation, induce sequential tipping of sea-ice cover and the ocean's meridional overturning circulation. The ocean component, represented by the well-known Stommel box model, is shown to display so-called rate-induced tipping. Here, an abrupt resurgence of the overturning circulation is induced before a bifurcation point is reached due to the fast rate of change of the sea-ice. During the rate-induced transition, the system is attracted by the stable manifold of a saddle. This results in a significant delay of the tipping if the system spends longer periods of time in the vicinity of the saddle before escaping towards the alternative state of a vigorous overturning circulation. The delay opens up the possibility for an early warning of the impending abrupt transition by detecting the change in linear stability. We propose early warning by estimating properties of the Jacobian from the noisy time series, which are shown to be useful as a generic precursor to detect rate-induced tipping.

## 1 Introduction

Multiple elements in the Earth system are believed to be at risk of undergoing abrupt and irreversible changes in response to rising atmospheric Greenhouse gas concentrations. Among others, the Arctic sea-ice, the Greenland and West Antarctic ice sheets, the Amazon rainforest and the Atlantic Meridional Overturning Circulation (AMOC) have been identified to potentially cross such tipping points at varying levels of global warming (Lenton et al., 2008). While an abrupt decline of the Arctic sea-ice is already well underway (IPCC, 2019), for a system like the AMOC it is much more uncertain if and when a tipping point will be reached. Nevertheless, it constitutes a risk that deserves attention as it has been observed across the hierarchy of climate models (Weijer et al., 2019), and there is evidence that it has occurred repeatedly during the last glacial period (Henry et al., 2016). Such past changes of the AMOC likely led to abrupt climate changes known as Dansgaard-Oeschger (DO) events (Dansgaard et al., 1993). These are the most significant instances of large-scale climate change in the past, but the underlying mechanisms remain debated.

Mathematically, tipping points are typically understood as a transition from one stable attractor of the system to another. Most often, this transition is associated with a bifurcation or attractor crisis, where a system state loses stability as a critical threshold in a control parameter is crossed, leading to tipping to another attractor (*bifurcation tipping*). However, tipping can occur also before a critical threshold is crossed. Stochastic perturbations may induce a transition to an alternative attractor (*noise-induced tipping*). Furthermore, some systems can fail to track their moving equilibrium and tip to another attractor



while no bifurcation was crossed, given there is a change in a control parameter at a high enough rate (*rate-induced tipping*) (Wieczorek et al., 2011; Ashwin et al., 2012).

Rate-induced transitions can be expected to play a role when there is a time scale separation in interacting, coupled systems. Here, changes in one subsystem alter the conditions of another and act as a rapidly changing control parameter that could cause a rate-induced transition. Such transitions might occur in the real climate system, where a vast range of time scales is represented by atmosphere, ocean and cryosphere, and where important climate parameters, such as ice melt in the polar regions, currently display accelerating rates of change (Trusel et al., 2018; Bevis et al., 2019; The IMBIE Team, 2020). Indeed, a rate-induced collapse of the AMOC has been shown recently in a global ocean model (Lohmann and Ditlevsen, 2021).

Rate-induced transitions in coupled systems become even more likely if one of the subsystems experiences abrupt change due to tipping. This constitutes a cascade of subsequent tipping points. Cascades of tipping points in coupled ecological or climate models have been considered before (Cai et al., 2016; Dekker et al., 2018; Rocha, J. C. and Peterson, G. and Bodin, Ö and Levin, S., 2018; Klose et al., 2020; Wunderling et al., 2020). However, cascading transitions where subsystems permit rate-induced tipping have not been studied yet.

Here we explore such a mechanism with a conceptual sea-ice-ocean model. The model describes the influence of changing polar sea-ice cover on the AMOC and features the possibility of a rate-induced resurgence of the AMOC. While an AMOC resurgence is not relevant for contemporary climate change, it plays an important role in past abrupt climate changes and DO events in particular, where it is thought to be responsible for the transitions from cold (so-called *stadial*) conditions to prolonged episodes of milder (*interstadial*) conditions during the last glacial period. It is still unknown what drove these transitions and

the associated resurgences of the AMOC. In climate models, an abrupt collapse of the AMOC can be induced by sudden discharges of freshwater into the North Atlantic, which is a phenomenon known to occur in the past (Heinrich, 1988). Similar events of sudden 'removal' of freshwater that potentially lead to an abrupt resurgence of the AMOC are less well-known. Instead, we consider changes in atmosphere-ocean heat exchange that could result from abrupt changes in sea-ice cover, which in turn could be driven by changing atmospheric wind stress. The potential of rapid sea-ice changes to advance the abrupt DO

warming events is well established (Li et al., 2005; Dokken et al., 2013; Vettoretti and Peltier, 2016; Sadatzki et al., 2019). The mechanism proposed here is different in that it involves a cascade: a tipping of the sea-ice cover leading to a rate-induced tipping of the ocean circulation due to the rapid increase in ocean heat loss.

Several lines of evidence from proxy data and climate model simulations motivate such a sequence of events. Zhang and co-workers showed model simulations with abrupt climate changes similar to DO events by gradually varying the Northern

Hemisphere ice sheet topography, which led to shifts in the atmospheric circulation that altered the wind-driven export of sea-ice (Zhang et al., 2014). This eventually led to an abrupt decrease in North Atlantic sea-ice cover and a resurgence of the AMOC. Kleppin and co-workers reported spontaneous transitions of the AMOC that were triggered by the stochastic atmospheric forcing and subsequent changes in North Atlantic sea-ice (Kleppin et al., 2015). Ice core data indicate that abrupt shifts in the sea-ice extent at the onset of DO events were preceded by shifts in atmospheric circulation by about a decade

(Erhardt et al., 2019). Furthermore, there is evidence for gradual trends leading up to the abrupt shifts in both sea-ice and





atmospheric circulation proxies, indicating an underlying parameter shift that might be mutually expressed in sea-ice and atmosphere (Lohmann, 2019; Sadatzki et al., 2019).

The conceptual model proposed here furthermore gives some interesting insight into dynamical phenomena in systems combining time-dependent and stochastic forcing. We find that the ocean component of our model (the well-known Stommel box model) displays rate-induced tipping in what could be called a 'soft' tipping point. Here, due to a non-smooth bifurcation, tipping occurs always before the bifurcation point is reached, even if the rate of change in the parameter shift is arbitrarily slow. Rate-induced tipping involves attraction by the stable manifold of a saddle, which can lead to a significant delay of the tipping under stochastic forcing. We also propose an early warning indicator to detect rate-induced tipping; so far only early warning signals specific to bifurcation tipping are known (Held and Kleinen, 2004; Dakos et al., 2008; Scheffer et al., 2009, 2012).

The paper is structured as follows. In Sec. 2 the coupled conceptual model is presented. We then show rate-induced tipping of the ocean component (the Stommel box model) in the deterministic and stochastic case in Sections 3.1 and 3.2, respectively. Thereafter, the cascading dynamics of the coupled model are presented (Sec. 3.3). Early-warning signals for the cascade, as well as for the rate-induced tipping, are investigated in Sec. 3.4 and Sec. 3.5. The results are discussed in Sec. 4, and our conclusions are given in Sec. 5.

## 2 Model

### 2.1 Ocean component: Stommel's '61 box model

We consider the Stommel box model of the Atlantic thermohaline circulation (Stommel, 1961), with added noise to represent variations in the atmospheric forcing on very short time scales. The model describes the overturning flow $\psi$ in between well-mixed polar and equatorial ocean basins as proportional to the density difference

$$\psi \propto (\rho_p - \rho_e) = [\alpha_T(T_e - T_p) - \alpha_S(S_e - S_p)], \tag{1}$$

where the density is given by the equation of state of seawater

$$\rho_{e,p} = \rho_0 \left[1 - \alpha_T(T_{e,p} - T_0) + \alpha_S(S_{e,p} - S_0)\right], \tag{2}$$

with reference densities, temperatures and salinities $\rho_0$, $T_0$ and $S_0$, respectively. The two model variables represent the dimensionless temperature difference $T = \alpha_T(T_e - T_p)$ and salinity difference $S = \alpha_S(S_e - S_p)$ in between the boxes. This defines the dimensionless overturning strength

$$q = T - S. \tag{3}$$

The temperature and salinity in the boxes relax towards an atmospheric temperature and salinity forcing $T_{e,p}^a$ and $S_{e,p}^a$. The meridional difference of the forcing drives the circulation and is represented by the two parameters $\eta_1 \propto (T_e^a - T_p^a)$ and $\eta_2 \propto (S_e^a - S_p^a)$. A third parameter represents the time scale ratio of the temperature and salinity relaxation $\eta_3 = \frac{\tau_T}{\tau_S}$. The model is





then defined by the stochastic differential equations

$$dT_t = (\eta_1 - T - |T - S|T)\,dt + \sigma_T dW_{T,t}$$
$$dS_t = (\eta_2 - \eta_3 S - |T - S|S)\,dt + \sigma_S dW_{S,t}, \tag{4}$$

with the Wiener processes $W_{S,t}$ and $W_{T,t}$. Time is scaled with respect to the ocean time-scale $\tau_T = 200$ years. For a more detailed derivation of the model see Dijkstra (2008). Over large regions of parameter space the deterministic system has two stable equilibria, which are referred to as the circulation 'on' and 'off' states. For the 'on' state we have $T > S$, where

the temperature forcing gradient dominates the opposing salinity forcing gradient and drives the circulation. The 'off' state ($S > T$) corresponds to a reversed circulation, which is weaker and dominated by the salinity forcing gradient. In Fig. 1a-b we show deterministic bifurcation diagrams of $q$ with respect to the parameters $\eta_1$ and $\eta_2$. In both cases, the 'on' state loses stability in a regular saddle-node bifurcation, whereas the 'off' state destabilizes in a non-smooth saddle-node bifurcation. The existence and extent of bi-stability depends on the parameter $\eta_3$. A large time scale separation (slower salinity damping)

leads to a large region of bi-stability, whereas as the salinity damping approaches the time scale of temperature damping, the bistability disappears (Fig. 1c). This is because a faster salinity damping disables the positive salt advection feedback, which gives rise to the bi-stability.

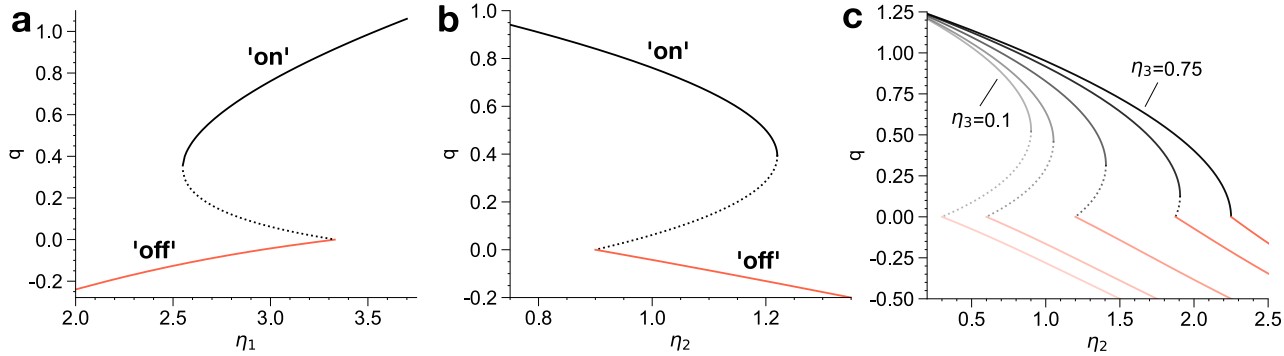

**Figure 1.** **(a)** Bifurcation diagram of the Stommel box model with $\eta_1$ as control parameter, $\eta_2 = 1.0$ and $\eta_3 = 0.3$. Solid lines indicate branches of stable fixed points, whereas dotted lines indicate unstable fixed, or saddle, points. **(b)** Bifurcation diagram with $\eta_2$ as control parameter, $\eta_1 = 3.0$. **(c)** Dependence of bi-stability on $\eta_3$. The individual bifurcation diagrams with $\eta_2$ as control parameter are shown with decreasing bistability interval as $\eta_3$ is increased from 0.1 up to 0.75.

## 2.2   Coupled sea-ice-ocean model

The ocean model is coupled to a sea-ice component in the polar ocean box, which is a modified version of the energy-balance

model described in Eisenman and Wettlaufer (2009) and Eisenman (2012), where the seasonal cycle and effects of the sea-ice thickness are disregarded. The changing sea-ice cover acts to insulate the polar ocean to varying degrees from the cold atmospheric temperature forcing $T_p^a$, thus modulating the temperature forcing gradient $\eta_1 \propto (T_e^a - T_p^a)$. A schematic of the





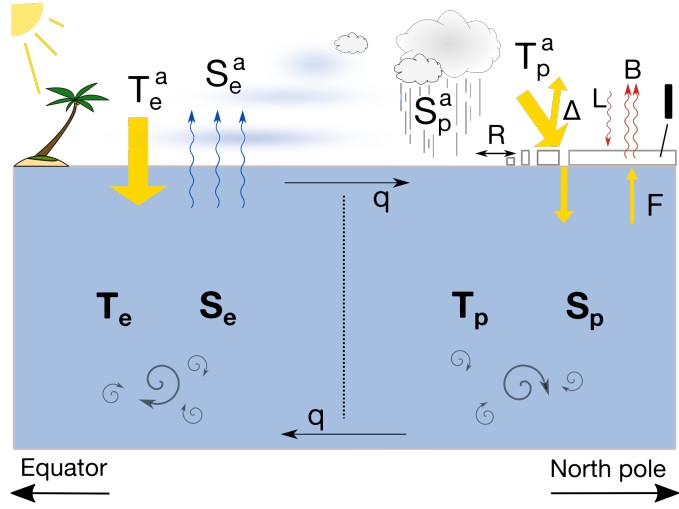

**Figure 2.** Schematic of the coupled sea-ice-ocean model including model parameters and variables (bold). A description of the parameters is given in Tab. 1. The well-mixed polar and equatorial ocean boxes are coupled by a surface flow $q$, along with an identical return flow at the bottom. The ocean component is reduced to the two variables $T \propto T_e - T_p$ and $S \propto S_e - S_p$. In the polar ocean box, the sea-ice cover $I$ insulates the ocean from the cold atmospheric temperature $T_p^a$.

coupled model including model variables and important parameters is given in Fig. 2. The deterministic sea-ice component is defined (Eisenman and Wettlaufer, 2009) by

$$\frac{dI}{dt} = \Delta \tanh\left(\frac{I}{h}\right) + [R_0\Theta(I) - B]\,I + L - F - 1 + R, \tag{5}$$


with the Heaviside step function $\Theta(\cdot)$. Time is scaled with respect to $\tau_I = 1$ year. The parameters and their values are described in Tab. 1. While $I > 0$ corresponds to a positive sea-ice cover, $I < 0$ represents zero sea-ice cover and the variable instead is a measure of the enthalpy of the surface ocean (Eisenman, 2012). The control parameter $R$ models influences on the sea-ice concentration due to external factors, such as export or import of sea-ice into the North Atlantic via changes in wind stress.

While in the climate system $R$ is driven by slower processes, such as changes in ice sheet topography, we treat it as a control parameter. We use parameter values from Eisenman (2012), which yield a sea-ice component that is bi-stable with respect to $R$. As seen in Fig. 3, for a range of $R$ there exists a stable state with a positive sea-ice cover (red), as well as a state with zero sea-ice cover $I < 0$ (black). This range is bounded by two saddle-node bifurcations. The stable state with sea-ice cover collapses at $R = -0.282$. We define the state at $R = 0$ as the stadial state, yielding a fixed point with positive sea-ice cover

$I_0^+ \approx 1.156$. A value of $h$ larger than given in Eisenman (2012) is used. This gives a smoother transition from an ice-free to an ice-covered state, and accordingly a smoother bifurcation diagram.

To capture transitions from stadial to interstadial conditions in the coupled sea-ice-ocean model, we consider the following mechanism. The polar ocean is insulated by a high sea-ice concentration from the atmospheric temperature forcing, preventing it from losing heat efficiently. As the sea-ice concentration decreases, the polar ocean becomes more and more exposed to





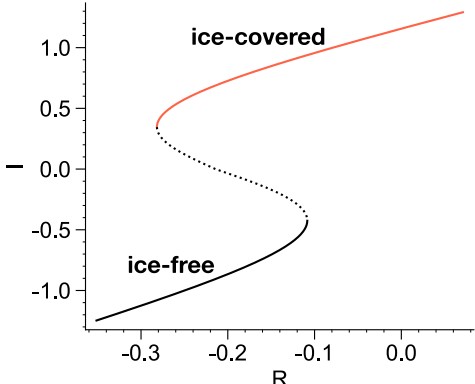

**Figure 3.** Bifurcation diagram of the sea-ice component for parameter values as in Tab. 1. The solid (dotted) lines indicate stable (unstable) fixed points.

the cold atmosphere and loses heat. Thus, the sea-ice variable modulates the parameter $\eta_1$. It is then defined as $\eta_1(I) = \eta_1^0 - \kappa\Theta(I)I$, again with the Heaviside function $\Theta(\cdot)$ since $I < 0$ corresponds to zero sea-ice cover. Together with additive noise to model fast atmospheric perturbations (Wiener process $W_{I,t}$), this yields the following coupled model:

$$dI_t = \left(\Delta\tanh\left(\frac{I}{h}\right) + [R_0\Theta(I) - B]I + L - F - 1 + R\right)dt + \sigma_I dW_{I,t}$$

$$\frac{\tau_T}{\tau_I}dT_t = \left(\eta_1^0 - \kappa\Theta(I)\cdot I - T - |T - S|T\right)dt + \sigma_T dW_{T,t} \tag{6}$$

$$\frac{\tau_T}{\tau_I}dS_t = \left(\eta_2 - \eta_3 S - |T - S|S\right)dt + \sigma_S dW_{S,t}$$

The value of $\kappa$ reflects the change in atmospheric temperature forcing when removing the sea-ice cover. In this conceptual

model it can only be chosen heuristically. We can for instance assume $\eta_1^0 = 3.0$ for an open ocean, and atmospheric temperature forcings in a glacial climate of 20 $^o$C and -10 $^o$C in the equatorial and polar box, respectively. Full sea-ice cover would limit the polar temperature forcing to 0 $^o$C and correspond to $\eta_1 = 2.0$. Even if the glacial polar atmosphere were above 0 $^o$C, given that it was colder than the surface ocean, extensive sea-ice cover would severely reduce heat loss to the atmosphere and thus effectively reduce $\eta_1$. Here we choose a scenario where during the stadial the sea-ice reduces the atmospheric forcing from

$\eta_1 = 3.0$ to $\eta_1 = 2.65$. $\kappa$ is then chosen such that $\eta_1 = 2.65$ at the stadial fixed point $I_0^+$ and $\eta_1 = 3.0$ for $I < 0$, yielding $\kappa = 0.35/I_0^+$. As a result, the ocean component is in the bi-stable regime for both full and zero sea-ice cover. A transition from stadial to interstadial will then be captured by decreasing $R$ from zero beyond the bifurcation point which tips the sea-ice component towards a state of $I < 0$, while the ocean remains in the bi-stable regime.





**Table 1.** Description of Model Parameters

| Parameter | Description | Value |
|---|---|---|
| $\eta_2$ | Salinity forcing gradient | 1.0 |
| $\eta_3$ | Temperature-salinity time scale ratio | 0.3 |
| $\kappa$ | Sea ice - ocean coupling | 0.303 |
| $\tau_T$ | Ocean time scale | 200 |
| $\tau_I$ | Sea ice time scale | 1.0 |
| $\Delta$ | Ocean - sea-ice albedo diff. | 0.43 |
| $h$ | Albedo transition smoothness | 0.5 |
| $R_0$ | Sea ice export | -0.1 |
| $B$ | Outgoing longwave radiation coeff. | 0.45 |
| $L$ | Incoming longwave radiation | 1.25 |
| $F$ | Ocean forcing on sea-ice | 1/28 |

## 3 Results

### 3.1 Rate-induced tipping and soft tipping points in Stommel model

In this Section, we investigate the tipping dynamics in the ocean component in the deterministic limit ($\sigma_T = \sigma_S = 0$). As noted above, there are non-smooth bifurcations in the Stommel model as the 'off' state loses stability, which leads to a resurgence of the AMOC. Leading up to these non-smooth bifurcations, the stable fixed points move in the same direction as the saddle point in terms of $T$ and $S$ when varying the control parameter. Thus, in a sufficiently transient parameter change, the saddle point can outpace the system state, which is trying to follow the moving equilibrium. This is illustrated in Fig. 4, where instantaneous parameter shifts and the corresponding movements of the system state vector in the bifurcation diagrams are indicated. When the saddle point moves past the system state, the system will tip towards the alternative stable state, which is the 'on' circulation in our case. Thus, tipping can occur even before the bifurcations points are reached, which is known as rate-induced tipping. While in the Stommel model this can happen for both $\eta_1$ and $\eta_2$ as control parameter, the behavior is more pronounced when changing $\eta_1$.

More rigorously, one has to consider the movement of the basin boundary as the control parameter is changed. The basin boundary is the stable manifold of the saddle, and it separates the basins of attractions of the 'on' and 'off' states, i.e. the sets of initial conditions that converge to the respective attractors. In Fig. 5, we illustrate the movement of the fixed point and basin boundary as $\eta_1$ is changed from 2.65 to 3.0. This corresponds to the scenario of a transition from stadial to interstadial sea-ice cover in the coupled model, as described in Sec. 2.2. Figure 5b shows that the 'off' fixed point before the parameter shift (open circle) lies inside the basin of attraction of the 'on' fixed point after parameter shift (blue area). This is a sufficient condition for rate-induced tipping, which has been called basin instability (O'Keeffe and Wieczorek, 2020), since for an instantaneous parameter shift, the system would tip to the other attractor. Similarly, as the system tries to follow the moving fixed point during a sufficiently fast parameter shift, it will fail to reach the 'off' basin (orange area) at the end of the parameter shift and tip to the 'on' fixed point. This happens for the blue trajectory, where the parameter is ramped up linearly within 300 years. In contrast,



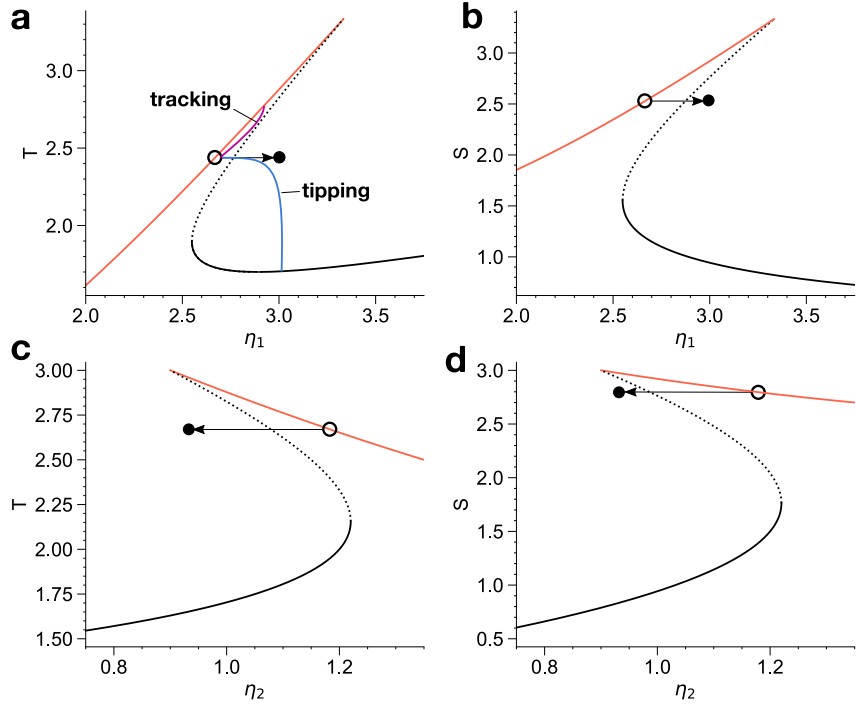

**Figure 4. (a,b)** Bifurcation diagram of the Stommel model fixed points in terms of the variables $T$ **(a)** and $S$ **(b)** as a function of $\eta_1$ as control parameter with $\eta_2 = 1.0$ and $\eta_3 = 0.3$. **(c,d)** Same, but with $\eta_2$ as control parameter and $\eta_1 = 3.0$ and $\eta_3 = 0.3$. Solid lines indicate stable fixed points, whereas dotted lines indicate saddle points. The horizontal arrows indicate the movement of the system state as the control parameter is changed instantaneously within the bi-stable regime. In **(a)** we illustrate how the system state may track the moving equilibrium for a slow parameter shift (purple trajectory), or tip to the undesired equilibrium in a fast parameter change (blue trajectory).

the purple trajectory shows that tipping does not occur for a ramping duration of 500 years. For this given size of the parameter shift, there is a critical rate of parameter change in between these two values.

Figure 6 shows time series of $q$ for simulations with different ramping durations. The realizations in a) and b) tip to the 'on' attractor, while the realizations in panels c) and d) track the moving 'off' equilibrium. The critical ramping duration is in

between the 388.5 and 390 years employed in panels b) and c). Comparing a) to b), one observes a delay in the tipping in b) of multiple thousand years. This occurs because for close-to-critical rates, the system state passes by very closely to the saddle point, where it remains for a long time as the dynamics slows down before being ejected. The reason for the close approach of the saddle is that the system state is being attracted to the saddle by its stable manifold, which is also the basin boundary. If one were to use the exact critical ramping duration, the system state would evolve precisely towards the saddle and remain there.

Such trajectories are called maximum canards (O'Keeffe and Wieczorek, 2020). This behavior is also seen for trajectories that eventually track the moving equilibrium, as in panel c). It is worth noting that the attraction by the stable manifold of the saddle continues after the parameter shift is already over, as shown in the inset in panel c.





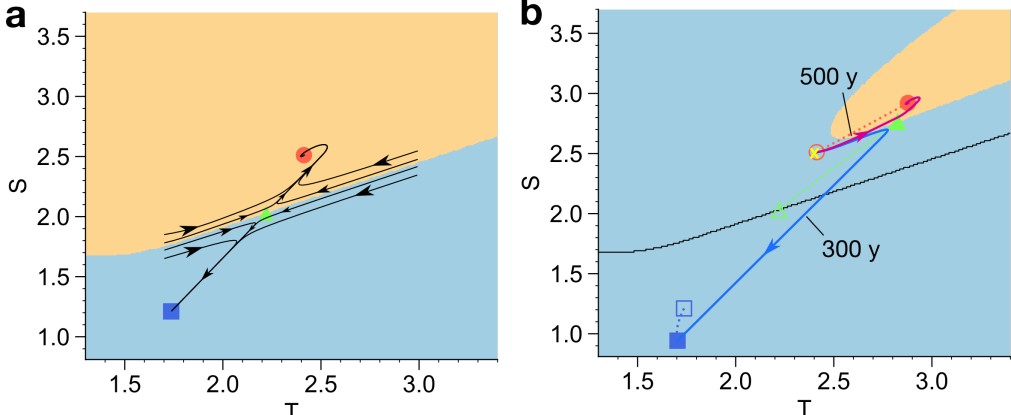

**Figure 5.** Phase portraits of the Stommel model with basins of attraction and fixed points. Squares and dots indicate stable fixed points, and triangles denote saddle points. **(a)** Phase portrait for $\eta_1 = 2.65$ with several flow lines to indicate the dynamics around the saddle. The basin of attraction of the 'off' ('on') state is shaded in orange (blue). **(b)** Phase portrait for $\eta_1 = 3.0$. Two trajectories, where $\eta_1$ is ramped linearly from $\eta_1 = 2.65$ to $\eta_1 = 3.0$ within 300 and 500 years are shown in blue and purple, respectively. The initial conditions $T, S = (2.4, 2.5)$ are indicated by the yellow cross. Open symbols indicate the positions of the fixed points at $\eta_1 = 2.65$, and the black curve indicates the corresponding basin boundary from **(a)**.

The critical ramping duration depends on the amplitude $\Delta\eta_1$ of the parameter shift (Fig. 7a). Rate-induced tipping becomes possible at a certain $\Delta\eta_1$, where the basin instability condition is first satisfied. Increasing $\Delta\eta_1$ then leads to a very rapid
increase of the critical ramping duration $D_c$. Thereafter, $D_c$ keeps increasing and actually diverges as the bifurcation is approached. This is due to the non-smoothness of the bifurcation. Here the basin boundary gets close to the attractor very quickly as the bifurcation is approached. This is seen by the super-linear scaling of the shortest distance to the basin boundary in Fig. 7b, where it is compared to the square root scaling of the smooth bifurcation in the sea-ice component. As the bifurcation is approached, the basin boundary gets arbitrarily close to the attractor, and even very small and slow parameter increases lead
to tipping. Thus, the non-smooth bifurcation leads to a 'soft' tipping point: In practice, there is no hard critical threshold of the parameter, but for any parameter shift at finite rate, the tipping will occur earlier and the location of the tipping point will depend on the trajectory of the parameter shift.

## 3.2   Noisy rate-induced tipping

We now consider added noise in the ocean component, which models variations in atmospheric forcing on very short time
scales. In addition to the 'soft' tipping just described, the stochastic perturbations further blur the critical threshold leading to tipping. For a given amplitude of the parameter shift, there is no longer a critical rate, but a range of rates where the probability of tipping goes from 0 to 1. Figure 8a shows how this range of rates expands for increasing noise level. Note that since the system features unbounded noise, here we consider finite time tipping probabilities during a simulation time of 5000 years.





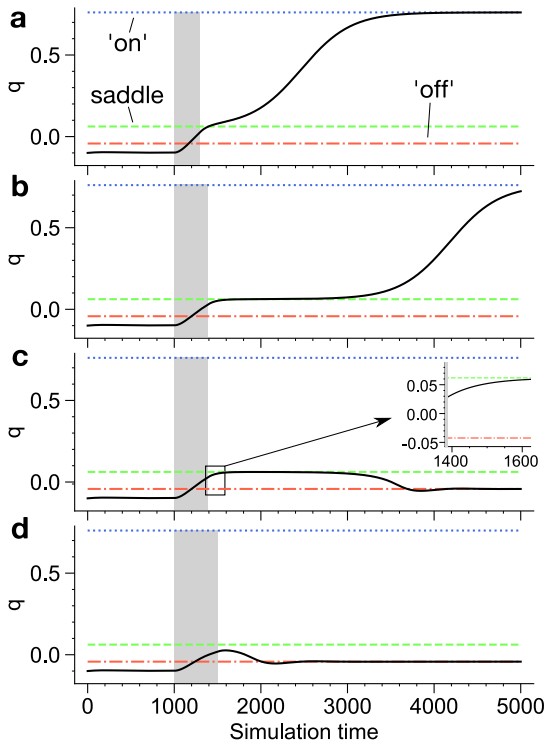

**Figure 6.** Time series of $q = T - S$ in the Stommel model when ramping the parameter from $\eta_1 = 2.65$ to $\eta_1 = 3.0$ at different rates. The realizations are initialized at $T, S = (2.4, 2.5)$, which is close to the 'off' fixed point at $\eta_1 = 2.65$. The duration of the ramping is indicated by the gray shading. The realizations in **(a)** and **(b)** with ramping durations of 300 and 388.5 years, respectively, tip from the 'off' to the 'on' attractor. The realizations in **(c)** and **(d)** with ramping durations of 389 and 500 years, respectively, track the moving 'off' attractor. The 'on', 'off' and saddle fixed points at $\eta_1 = 3.0$ are shown as horizontal lines.

Eventually there will be a noise-induced transition to the 'on' attractor, especially from the 'off' attractor at $\eta_1 = 3.0$ for higher
noise levels.

By introducing noise, tipping becomes a mixture of rate-induced and noise-induced transitions, since the unbounded noise allows the system to cross the basin boundaries of the deterministic system in any circumstances. Still, for low noise levels the behavior strongly resembles the deterministic case. As discussed earlier, for a ramping speed relatively close to the critical rate, the tipping involves an escape from the saddle. This behavior is robust for low noise levels, where the stochastic fluctuations
cannot overcome the attraction of the stable manifold of the saddle. Thus, the system approaches the saddle, before being ejected from its vicinity.

As the noise level is increased, there are noise-induced early tippings as well as significantly delayed tippings. In order to quantify when a tipping is 'early' or 'late', we need to define the moment when the system actually tips. For the deterministic system, a sensible choice would be the time when the moving, quasi-stationary basin boundary is crossed, since this is the first
moment that the system would tip in case the parameter shift would be stopped suddenly. However, for the noisy system this





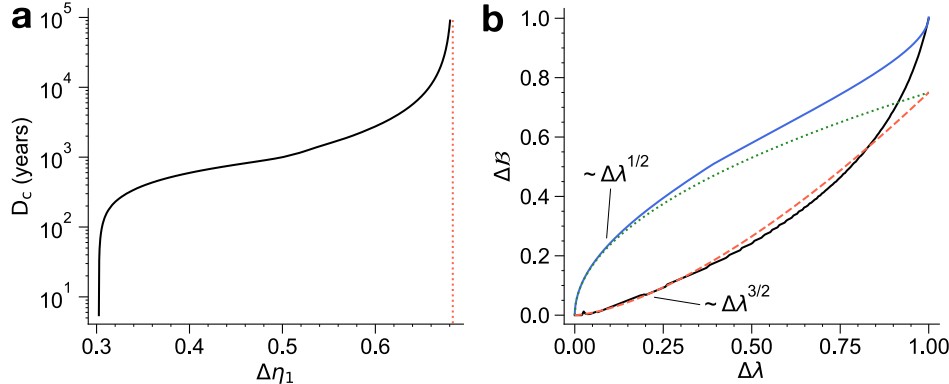

**Figure 7. (a)** Critical ramping duration beyond which there is a rate-induced tipping in the Stommel model when shifting the parameter from $\eta_1 = 2.65$ to $\eta_1 = 2.65 + \Delta\eta_1$. The 'off' attractor loses stability in the bifurcation at $\eta_1^{\text{off}} = 3.333$, as indicated by the red dashed line. **(b)** Normalized shortest distance to the basin boundary $\Delta\mathcal{B}$ as a function of the normalized distance to the bifurcation $\Delta\lambda = (\eta_1^{\text{off}} - \eta_1) \cdot (\eta_1^{\text{off}} - \eta_1^{\text{on}})^{-1}$. $\eta_1^{\text{on}}$ is the parameter value at the other saddle node bifurcation of the 'on' state. The black, solid curve shows the results of the Stommel model, and a proposed super-linear scaling is shown by the dashed curve. Also shown are results for the smooth bifurcation in the sea-ice component (blue solid) and the corresponding square root scaling (dotted).

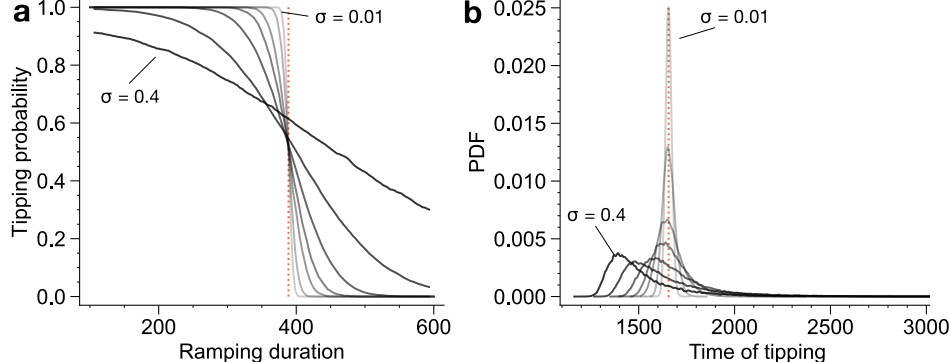

**Figure 8. (a)** Probability of a rate-induced tipping in the Stommel model from the 'off' to the 'on' state as a function of the linear parameter ramping duration from $\eta_1 = 2.65$ to $\eta_1 = 3.00$. Different noise levels $\sigma_T = \sigma_S = \sigma$ are considered: $\sigma = 0.01$ (lightest gray curve), $\sigma = 0.02$, $\sigma = 0.04$, $\sigma = 0.06$, $\sigma = 0.1$, $\sigma = 0.2$ and $\sigma = 0.4$ (darkest gray curve). The red dashed line is the critical ramping duration in the deterministic system. **(b)** Probability distributions of the time of tipping, defined by the first crossing of $q > 0.1$, for different noise levels. The ramping is started in year 1000 and the duration is fixed at 300 years. The red dashed line is the time of tipping in the deterministic system.

does not guarantee tipping, since the system may cross back to the other basin at any time. As a heuristic definition of tipping, we can instead detect the departure from the vicinity of the saddle in terms of the overturning $q$, as the tipping is associated





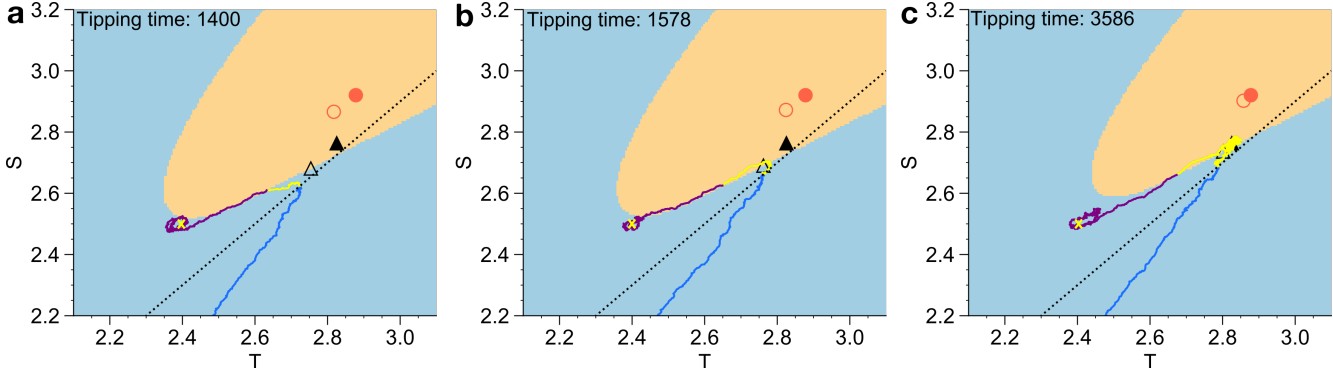

**Figure 9. (a,b,c)** Three realizations in phase space of the Stommel model with $\sigma_T = \sigma_S = 0.2$, where $\eta_1$ is ramped from $\eta_1 = 2.65$ to $\eta_1 = 3.00$ over 300 years. The filled dot (triangle) marks the 'off' fixed point (saddle) at $\eta_1 = 3.0$. The colored areas are the quasi-stationary basins of attraction at the time when their boundary is first crossed. The colored basins of attractions are given at the time of first basin crossing of the trajectories, which change color from purple to yellow. The initial conditions $T, S = (2.4, 2.5)$ are indicated by the yellow cross. The locations of the saddle point (triangle) and the 'on' fixed point at this time are shown with open symbols. The threshold $q = T - S = 0.1$ used to define the time of tipping is shown as the dotted line.

with a monotonic increase of $q$ (see Fig. 6). Thus, as tipping we define the first crossing of $q = 0.1$, which is a slightly larger value than at the saddle to allow for some fluctuations around it. In phase space this defines a straight line.

Figure 9 shows the crossing of this threshold, as well as the moments when the basin boundary is crossed for three different realizations with a ramping duration of 300 years and $\sigma_T = \sigma_S = 0.2$. The time of tipping varies significantly and depends primarily on the proximity of the approach to the saddle and the subsequent time spent in its vicinity. Whereas Fig. 9b shows a realization with tipping close to the deterministic scenario, the realization in Fig. 9a) leaves the stable manifold early and does not approach the saddle closely. The realization in Fig. 9c approaches the saddle very closely and remains there for a long

period of time.

The tipping time distribution and its dependence on the noise level are shown in Fig. 8b. In our case of a ramping duration slightly below the critical value of the deterministic system, there are three regimes of noise levels. For low noise ($\sigma = 0.01$, $\sigma = 0.02$, $\sigma = 0.04$ and $\sigma = 0.06$ in Fig. 8b) the trajectories are very similar to the deterministic case, and it is very unlikely that the noise pushes the system closer to the saddle. Thus, the tipping time is distributed closely around the deterministic

value. For intermediate noise ($\sigma = 0.1$ and $\sigma = 0.2$ in Fig. 8b), some early noise-assisted tippings are possible, as seen by the shift of the mode of distribution to earlier tippings. These early tippings arise as the stochastic perturbations prevent the system from approaching the saddle as closely as in the deterministic case. Additionally, there is a good chance that the noise pushes the system closer to the saddle, where it can stay for a long time (multiple thousand years) as the dynamics slow down before escaping. This leads to long-tailed tipping time distributions. For larger noise ($\sigma = 0.4$ in Fig. 8b), even earlier noise-assisted

tippings are seen, as well as some delayed tippings. However, the latter occur not as frequently as for intermediate noise, since the residence time at the saddle is shorter.



### 3.3 Cascading dynamics

We now consider the coupled model and investigate how a stadial-interstadial transition can arise as a cascading tipping of the two components. The cascade is initiated by a change in the control parameter $R$ leading to a decrease and eventual tipping of the sea-ice to $I < 0$. Subsequently, the modulations of the parameter $\eta_1$ by a decrease of $I$ can be expected to induce a rate-induced resurgence of the AMOC. On the one hand, this is because the time scale of the sea-ice is much shorter and the dynamics are thus very fast as the sea-ice tips. On the other hand, even if the sea-ice does not change fast, when the amplitude of the change in $\eta_1$ becomes larger, there will be rate-induced tipping anyway due to the 'soft' tipping point in the Stommel model described earlier. As a result, we choose a scenario where the coupling $\kappa$ is such that the ocean model remains in the bi-stable regime and a rate-induced AMOC resurgence is the only pathway to tipping. As described in Sec. 2.2, this can be exemplified by a change in $\eta_1$ from $\eta_1 = 2.65$ at the stadial sea-ice fixed point for $R = 0$ to $\eta_1 = 3.0$ for a collapsed sea-ice cover $I < 0$. Simulations with these parameters are qualitatively representative for a wider range of coupling strengths and rates of changing $R$.

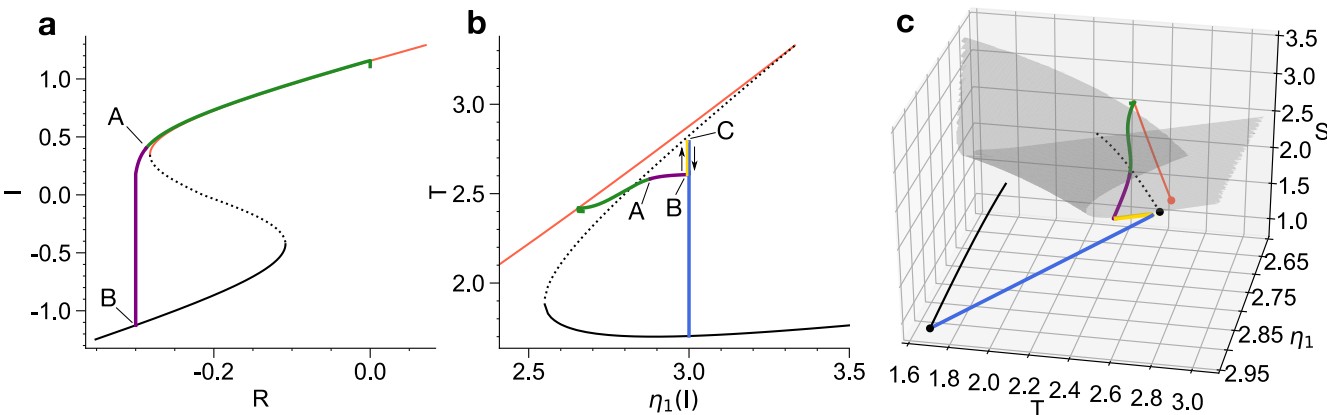

**Figure 10.** Cascading stadial-interstadial transition in the coupled sea-ice-ocean model where $R$ is ramped from $R = 0$ to $R = -0.3$ within 340 years and kept constant afterwards. **(a)** Trajectory of the sea-ice component as function of the control parameter $R$. **(b,c)** Trajectory of the ocean component as function of the changing parameter $\eta_1(I(t)) = \eta_1^0 - \kappa\Theta(I(t)) \cdot I(t)$. The tipping cascade consists of several steps separated by the points A, B, and C, and marked by different colors in the trajectories (see main text). The gray surface in **(c)** is the moving basin boundary corresponding to the changing $\eta_1(I(t))$.

Figure 10 shows trajectories for a cascading stadial-interstadial transition in the deterministic limit when $R$ is ramped down from $R = 0$ to $R = -0.3$ over 340 years, which has several stages. First, the sea-ice slowly decreases as $R$ is decreased and the ocean component tries to track the moving equilibrium (green segment of trajectories in Fig. 10). At point A, 325 years after the start of ramping, the sea-ice passes the bifurcation point and rapidly tips to $I < 0$ (purple segment in Fig. 10). This leads to a quick movement of $\eta_1(I)$ towards $\eta_1 = 3.0$, which is reached at point B, 350 years after the start of ramping (Fig. 10b). As a result, the ocean state crosses the moving basin boundary (gray surface in Fig. 10c) from above, and is thus determined to





undergo rate-induced tipping to the 'on' attractor (black solid curve). Before tipping, the ocean state is attracted by the stable manifold (i.e. the basin boundary) of the saddle (yellow segment). Finally, at point C (700 years after the start of ramping) the ocean component escapes the vicinity of the saddle and tips towards the 'on' state (blue segment).

There is a critical time scale below which such a cascading transition with a rate-induced tipping is possible. This is a combination of the rate of change in the control parameter $R$ and the speed of the tipping of the sea-ice, which will be kept

fixed here. As additive noise is included in the model, the boundary of tipping in terms of the ramping time of the control parameter is again blurred. Figure 11a shows the tipping probabilities for different noise levels as a function of the ramping time of $R$. The result is very similar to the ocean only case, except that because of the fast tipping in the sea-ice, the ramping times leading to tipping are slightly higher. The picture looks different as we increase the noise level in the sea-ice component, as seen in Fig. 11b. Here, the ramping times that yield significant tipping probabilities simply increase with the noise level

without a large simultaneous decrease of the tipping probability for lower ramping durations. This is because noise-induced transitions to $I < 0$ occur before the bifurcation of $I$ is crossed. Since these transitions happen on the fast sea-ice time scale, a rate-induced tipping of the ocean model becomes possible even when $R$ is changed more slowly. As in the ocean-only case, the tipping cascade involves a saddle escape, which can lead to significant tipping delays as noise forcing of intermediate strength is included. Next, we will discuss this in more detail and relate it to potential pre-cursor signals leading up to such transitions.

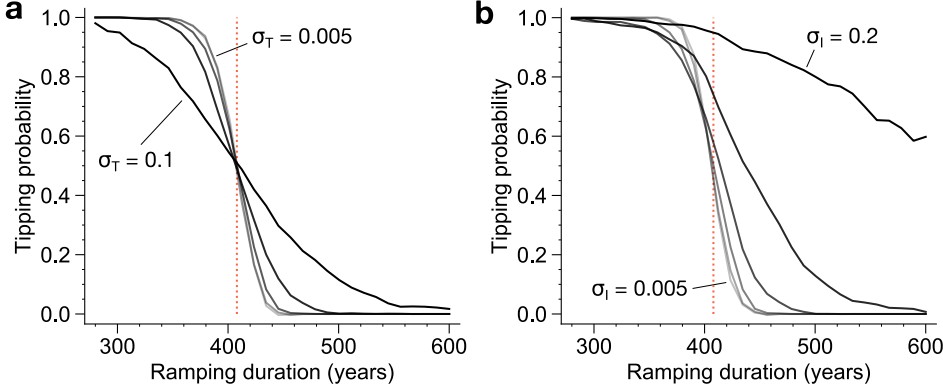

**Figure 11.** Probability of a cascading transition in the coupled sea-ice-ocean model when changing the control parameter $R$ linearly from $R = 0$ to $R = -0.3$ within different ramping times. **(a)** Fixed noise level $\sigma_I = 0.02$ in the sea-ice component and varying noise levels $\sigma_T = \sigma_S = \sigma = 0.005$ (lightest gray), $\sigma = 0.01$, $\sigma = 0.02$, $\sigma = 0.04$, and $\sigma = 0.1$ (darkest gray) in the ocean component. **(b)** Fixed noise level $\sigma_T = \sigma_S = 0.02$ in the ocean component and varying noise levels $\sigma_I = 0.005$ (lightest gray), $\sigma_I = 0.01$, $\sigma_I = 0.02$, $\sigma_I = 0.04$, $\sigma_I = 0.08$, and $\sigma_I = 0.2$ (darkest gray) in the sea-ice component.

## 3.4   Early warning of the tipping cascade

Due to their irreversible nature, it is important to foresee impending tipping points using generic early warning signals that do not require detailed knowledge of the system dynamics. These are typically obtained from time series by estimating a





statistical indicator in a sliding window with appropriate detrending (see Sec. S1). For bifurcation tipping, a system often exhibits critical slowing down, which can be measured by increasing variance and autocorrelation. In Fig. 12 we show these

indicators estimated in a sliding window for the cascading transition in Fig. 10. As expected there is an increase in variance and autocorrelation of $I$ leading up to the bifurcation (Fig. 12c,d). Because of the speed of the parameter shift necessary to induce the cascade, the increases in the indicators do not fully exceed the variability prior to the parameter shift, but could still provide early warning with a reasonable skill. Due to the coupling one might expect a signature of the sea-ice critical slowing down in the ocean component. This is not seen here (Fig. 12e,f), since increasing fluctuations due to the sea-ice are small compared

to the variability in the ocean component for the chosen $\sigma$. If no noise is added to the ocean variables, critical slowing down can be detected in $T$ or $S$. This might be an example of scenarios proposed in Rypdal (2016) andBoers (2018), where it is hypothesized that a bifurcation in the sea-ice system is detectable as increased variance in the high frequencies of ice-core data prior to DO events. Similarly, the increasing fluctuations in $I$ may influence the ocean subsystem in a more consistent way as the bifurcation is approached. This increases the crosscorrelation especially on longer time scales, which can be measured

with detrended crosscorrelation analysis (DCCA). This has been proposed as early warning indicator for cascading transitions (Dekker et al., 2018). The method is similar to detrended fluctuation analysis, but instead of scaling in the variance, it measures scaling of the covariance of two signals with increasing time scales (for details see Zebende (2011) or Dekker et al. (2018)). We can detect a slight increase on average in the DCCA exponent of $I$ and $T$ (Fig. 12e,f) for the transition in Fig. 10). However, the increase found in individual time series is not statistically significant. This is mainly due to the large variance of the DCCA

estimator.

### 3.5   Early warning of rate-induced tipping in the Stommel model

During the rate-induced transition of the ocean component there is an increase in the ensemble variance, as can be seen by the shadings in Fig. 12b. This increase, as well as a corresponding increase in ensemble autocorrelation, has been proposed before as early warning signal for rate-induced tipping (Ritchie and Sieber, 2016). However, it results from the large spread

in the amount of time spent by individual realizations at the saddle before tipping to the other attractor, as shown above. The fluctuations in individual realizations, as used for operational early warning, do not show an increase in variance and autocorrelation. This can be seen in Fig. 12e-f, where no increases in sliding window variance and autocorrelation accompany the increase in ensemble variance. For the estimation of variance and autocorrelation in a sliding window, a detrending of the time series is necessary, such that remaining trends in the residuals are not larger than the fluctuations themselves. For our

detrending method using cubic functions, the severity of detrending, and thus the ability to remove sharp changes in the signal trend, depends only on the window size (see Sec. S1 for more details). In order to remove the trend due to the parameter shift regarded here, a window size of no more than 200 years is required (Fig. S1).

Detrending inevitably removes some of the original fluctuations. To show that the previous result is not a consequence of too severe detrending, we extract segments of the time series where the system is in the vicinity of the saddle and there are no sharp

trends. The fluctuations around the saddle are then compared to those around the initial attractor. We define the vicinity of the saddle as time periods in the simulations where $q = 0.06$ is first crossed until $q = 0.1$ is first crossed (Fig. 13). We regard the





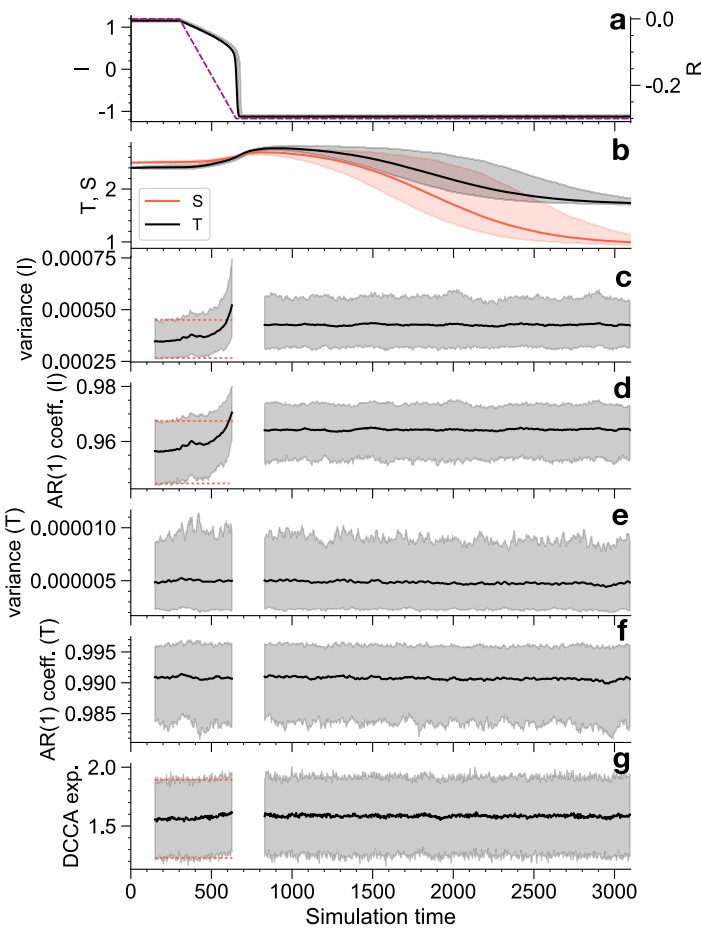

**Figure 12.** Ensemble simulations of the coupled sea-ice ocean model, where $R$ is ramped linearly from $R = 0$ to $R = -0.3$ within 350 years. **(a)** Time series of $R$ (dashed line) and mean time series of $I$ with a 90% confidence band of the ensemble (gray shading). **(b)** Mean time series and 90% confidence band of $T$ and $S$. **(c-f)** Indicators of critical slowing down for $I$ and $T$, estimated in a sliding window of 150 years, where the data in the window is detrended by a cubic function. The data is cut as the bifurcation in $I$ is crossed until after the last realization tips plus the sliding window length. **(g)** Detrended cross-correlation analysis (DCCA) exponent estimated from $I$ and $T$.

time series segments of an ensemble of realizations that stay in this vicinity for at least a certain duration. After detrending the segments by cubic functions, we calculate variance and autocorrelation yielding empirical distributions just before the moment of tipping. For each realization, we also choose a segment of the same duration taken just before the parameter shift starts, yielding distributions of variance and autocorrelation at the initial attractor. Figure 14 shows that variance and autocorrelation at the saddle are not increased, but actually slightly decreased compared to the initial attractor. This is seen best for longer segments (panels c and d) since the uncertainty in the estimators becomes smaller. In this case the average variance and autocorrelation is larger compared to panels a and b, since detrending the longer windows removes less variability.



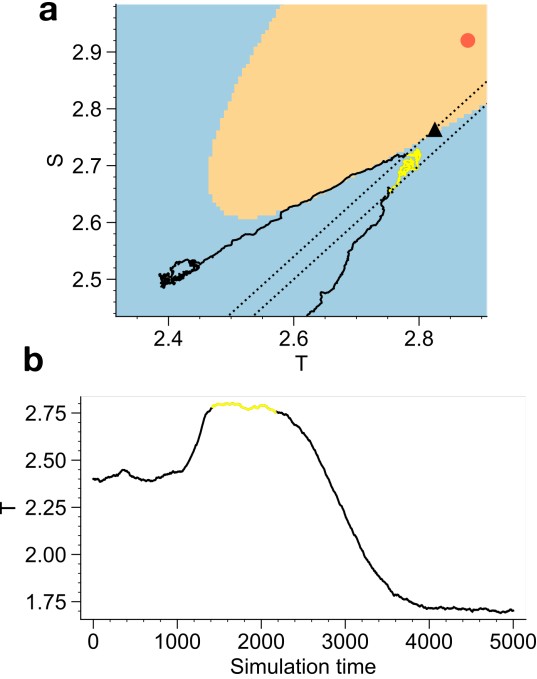

**Figure 13. (a)** Simulation in phase space of the Stommel model with $\sigma_T = \sigma_S = 0.2$, where $\eta_1$ is ramped from $\eta_1 = 2.65$ to $\eta_1 = 3.00$ within 300 years. The two dotted lines correspond to the levels $q = T - S = 0.06$ and $q = T - S = 0.1$. The trajectory in between the first crossing of these two thresholds is shown in yellow. **(b)** Corresponding time series of the variable $T$.

It thus does not appear that critical slowing down indicators apply to rate-induced tipping. Instead, we exploit that the system
is attracted towards the saddle where the dynamics are different to those at the initial attractor. If this difference can be detected before the system tips, a small perturbation in the right direction or a reversal of the parameter shift could push the system back in the desired basin of attraction. Saddles, which have at least one unstable direction in phase space, can be distinguished from attractors by a change from a negative to a positive real part of the largest eigenvalue of the Jacobian. Estimating the Jacobian from the time series in a sliding window could thus be a generic tool to detect the saddle escape involved in rate-induced
tipping, and we describe a method to do this in the Appendix A. With this method the elements of the Jacobian during rate-induced tipping of the Stommel model can be inferred and allow for the distinction of the dynamics around the different fixed points (Fig. S2). However, there are quantitative biases in the estimates of individual elements, and as a result the estimates of the real part of the largest eigenvalue in the vicinity of the saddle are not consistently positive. These biases could be a result of the detrending, of a too high noise level, or because the unstable dynamics are 'suppressed' since we consider realizations
before the escape from the saddle.

As a more reliable indicator we propose the actual elements of the Jacobian, since they are inferred in a qualitatively robust way (Fig. S2). This lowers the estimator variance compared to the eigenvalues, which are composed of the estimates of all elements. The off-diagonal elements record changes in sign of the feedbacks in between the system variables. Such changes



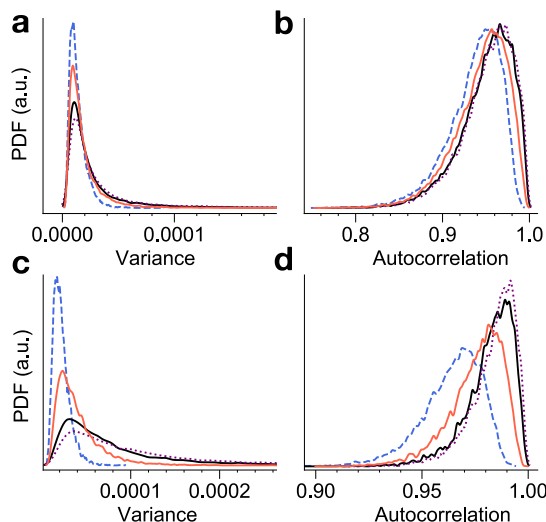

**Figure 14.** Distributions of variance and autocorrelation for ensembles of time series from the Stommel model ($\sigma_T = \sigma_S = 0.2$). These are estimated around the initial fixed point at $\eta_1 = 2.65$ (black) and close to the saddle point (orange, see main text), after detrending in time windows that correspond to the time period that the system spent in the vicinity of the saddle. **(a,b)** Results for realizations where these time windows were at least 300 years long. **(c,d)** Results for time windows of at least 700 years. Also shown are the distributions around the 'on' attractor (dashed) and the 'off' attractor at $\eta_1 = 3.0$ (dotted).

in feedback are common as a system move towards a saddle. We combine the off-diagonal elements to a scalar early warning
indicator $\mathcal{J}$, defined in Eq. A5. Figure 15a-f shows that $\mathcal{J}$ can distinguish the dynamics around the attractor (black) and the saddle (red) before tipping. The panels correspond to different lengths of the time windows used to estimate $\mathcal{J}$. The figure also shows probabilities $p$ of observing a value of $\mathcal{J}$ estimated around the attractor that is larger than a value of $\mathcal{J}$ in the vicinity of the saddle. This measures the performance of $\mathcal{J}$ as an early warning signal. For longer time windows, the distributions become better separated since the uncertainty of the estimator is reduced. While for longer time windows the performance is better
due to a reduced estimator variance, even for relatively short windows the indicator correctly identifies the departure from the attractor for most realizations.

An operational early warning signal can be constructed by estimating $\mathcal{J}$ in a sliding window, and raising an alert as soon as a threshold $\mathcal{J}_c$ is exceeded. The location of $\mathcal{J}_c$ relative to the tails of the distributions in Fig. 15a-f is a trade-off in between true and false positives (alerts). The performance of the alert as a binary classifier can be summarized in receiver operator characteristic (ROC) curves. The curve of a perfect classifier collapses to the point (0,1). Figure 15g shows that for realizations
that spend a longer time at the saddle, the indicator $\mathcal{J}$ comes close to a perfect classifier, detecting the saddle approach with very low false positive and very high true positive rates. Figure 16 shows $\mathcal{J}$ estimated from time series in a sliding window, along with critical slowing down indicators. $\mathcal{J}$ begins to rise sharply roughly 200 years after the ramping started and decreases slightly as most realizations leave the saddle towards the 'on' attractor. In contrast to the ensemble variance (orange), the





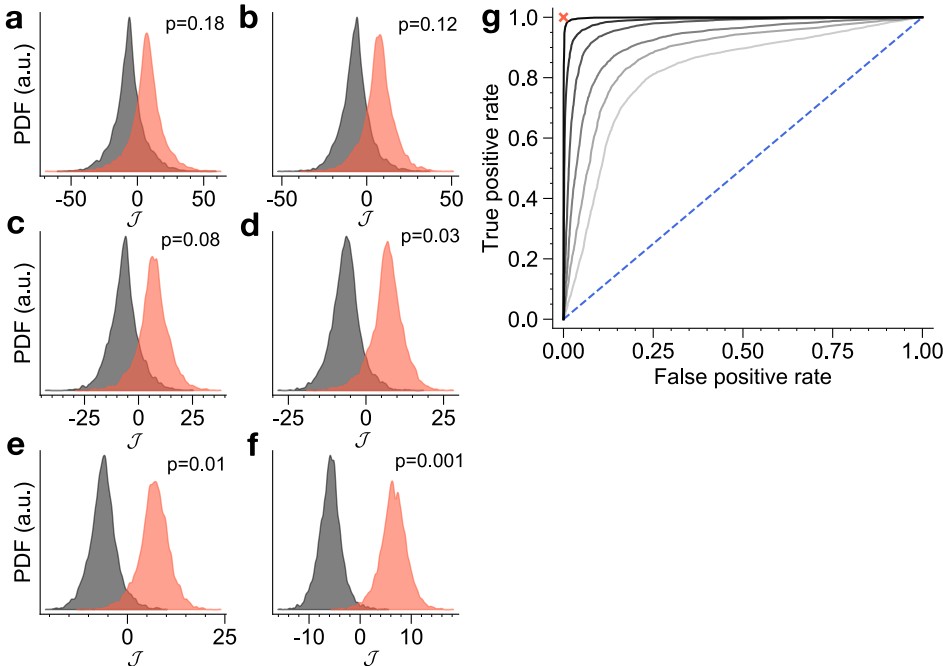

**Figure 15.** **(a-f)** Distributions of the early warning indicator $\mathcal{J}$ for ensembles of time series from the Stommel model ($\sigma_T = \sigma_S = 0.2$), estimated around the initial fixed point at $\eta_1 = 2.65$ (black) and close to the saddle point (orange). For each realization, $\mathcal{J}$ is estimated after detrending in a time window that corresponds to the time period that the system spent in the vicinity of the saddle. In increasing order, the panels show results for realizations where these time windows were at least 100, 150, 200, 300, 400 and 600 years long, respectively. **(g)** Receiver operator characteristic curves for the same time series ensembles, showing the false and true positive rates as the threshold $\mathcal{J}_c$ is increased from low (top right) to high values (bottom left). The increasing darkness in the gray scale of the curves corresponds to the increasing time window lengths, as above. The diagonal dashed line indicates the performance of a pure chance classifier. The red cross indicates a perfect classifier.

variance and autocorrelation in the sliding window show no signal, apart from a small artifact around the parameter shift, which is a remnant of imperfect detrending in the 200-year windows.

## 4 Discussion

In this work we propose a conceptual model describing a mechanism for abrupt climate change comprising a rate-induced resurgence of the AMOC as a response to increasing atmosphere-ocean heat exchange resulting from fast disappearance of

sea-ice. The latter occurs via a bifurcation tipping as a response to changing sea-ice export into the North Atlantic, which could be driven by changes in wind stress forcing. In the context of DO events, the proposed model merely describes the sequence of events leading to a stadial-interstadial transition. It can be easily extended to display self-sustained DO cycles by adding another slow variable that dynamically causes the parameter shift. It is thus compatible with both stochastic, externally forced



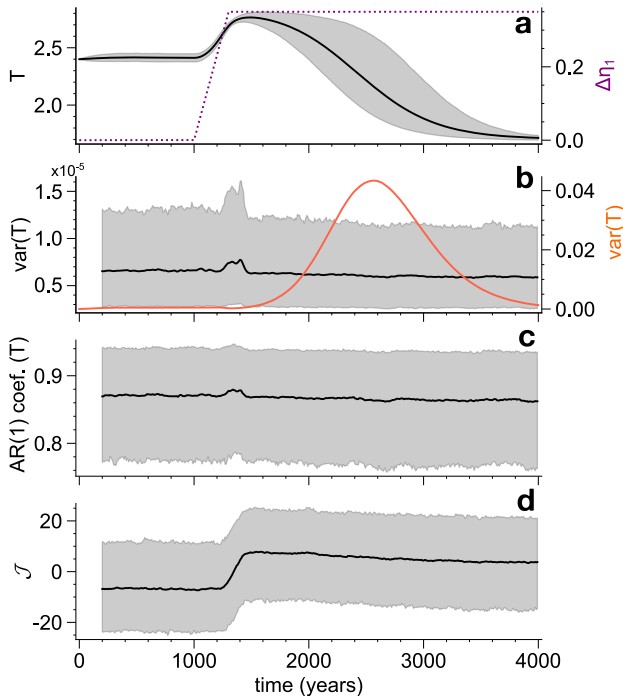

**Figure 16.** Early-warning indicators estimated in a 200-year sliding window from an ensemble of time series of the Stommel model, where $\eta_1$ is ramped from $\eta_1 = 2.65$ to $\eta_1 = 3.0$ within 300 years. **(a)** Time series of $T$ and the parameter ramp. **(b)** Variance estimated from the detrended time series, as well as the ensemble variance (orange). **(c)** Lag-1 autocorrelation in the sliding window. **(d)** Early-warning indicator $\mathcal{J}$ (Eq. A5) estimated from the Jacobian in the sliding window. Mean time series are shown in black and the range in between 5- and 95-percentiles are shaded in gray.

as well as self-sustained oscillatory dynamics driving DO cycles. Whether the proposed physical mechanism did indeed play

a role in past abrupt climate change needs to be tested with more complex models, as well as with analyses of new highly resolved and synchronized climate proxy records.

The type of cascade introduced here could be a common feature in coupled systems that feature multistability and time-scale separation. Here, a tipping in a fast subsystem can trigger a rate-induced transition of a slower subsystem even for weak coupling. Conversely, when there is no time scale separation but stronger coupling, the cascade can still occur in systems

with a non-smooth bifurcation. This is due to the 'soft' tipping point (Sec. 3.1), where the critical ramping duration to induce rate-induced tipping diverges as the parameter shift increases towards the bifurcation point. Consequently, we examined the mathematical details of the tipping cascade. The tipping occurs in several stages. During the parameter shift the ocean subsystem tries to track the moving equilibrium. As the sea-ice component tips, this fails and the system is instead attracted by the stable manifold of the saddle. The system then remains in the vicinity of the saddle as the dynamics slow down, before

escaping to the 'on' attractor. Adding noise leads to a broad distribution of the escape time towards the 'on' attractor. Both early tipping, where stochastic perturbations push the system away from the stable manifold, as well as significantly delayed



tipping is observed. In the latter case, noise pushes the system very close to the saddle, where it can get stuck for a very long time. A delay of rate-induced tipping for low noise levels has been reported for a one-dimensional gradient system (Ritchie and Sieber, 2016). In our system, due to the attraction by the stable manifold of the saddle, this is a robust feature seen for a
fairly large range of rates (both sub- and super-critical), as well as of noise levels. Thus, it opens up the possibility for early warning of rate-induced transitions.

Early-warning of the cascade before the initial tipping of the sea-ice is limited by the relatively fast parameter shift involved. Indicators proposed for cascading tipping points (Dekker et al., 2018) yield non-significant results. More research is needed to find better indicators that might rely on similar principles. Instead, we focused on the rate-induced tipping of the ocean
subsystem, since early warning signals for rate-induced tipping have not been developed. As in the case of fast passages through a bifurcation, for very fast parameter shifts one cannot hope for an early warning of rate-induced tipping. Here the system is not attracted by the saddle but evolves quickly towards the alternative attractor. However, for intermediate rates the tipping occurs via attraction towards a saddle. As the moving attractor is departed towards the saddle the linear stability changes. This are captured by the Jacobian matrix, which can be estimated from the time series. Here we propose to use
the off-diagonal elements of the Jacobian as early warning signal. These elements record changes in the sign of coupling in between the system variables, indicating a change in stability. The proposed indicator detects an approach of the saddle with significant skill, in particular for realizations where the system stays in the vicinity for a longer time, so that the Jacobian can be estimated with good precision. The actual tipping occurs by escaping the vicinity of the saddle, which is largely noise-induced. Thus, early warning in the sense of predicting the precise time of the saddle escape is hard to achieve. Early-warning
signals for saddle escapes have been proposed (Kuehn et al., 2015), but they require being very close to the saddle and very low noise. While the specific early warning signal proposed here may not apply to all cases of rate-induced tipping, the general procedure of detecting a qualitative change in the feedback structure of the system via the Jacobian or its eigenvalues should be widely applicable. For higher-dimensional systems early warning might even become easier, since there are often dominant eigenvalues and large differences in the effective dimensionality of the dynamics on the attractor versus the transient dynamics
during tipping. Other techniques for detecting transient dynamics might be useful here (Gottwald and Gugole, 2020). The phenomenology of cascading transitions involving rate-induced tipping that has been exemplified here is to be tested with models of different complexity in upcoming studies.

## 5   Conclusions

We propose a mechanism for abrupt climate change in a conceptual model that involves a cascade of tipping points: First,
the North Atlantic sea-ice cover collapses abruptly due to gradually changing external conditions. Subsequently, the AMOC resurges abruptly from a weak to a vigorous state in a rate-induced tipping, as a response to the fast rate of sea-ice decline enhancing the atmosphere-ocean heat exchange. Our analysis of the model illustrates that cascades of tipping points in weakly coupled climate subsystems with time-scale separation might be more likely than hitherto expected, given there are rate-dependent tipping points, or 'soft' tipping points associated with non-smooth bifurcations. While an early warning of tipping





points involving fast parameter shifts is generally difficult, we show that due to a delay in the tipping of the ocean circulation an estimation of the Jacobian can detect the impending abrupt transition. This may be applicable as generic early warning signal of rate-induced transitions.

**Appendix A:  An early warning indicator for rate-induced tipping**

We detect rate-induced tipping by identifying a departure from the initial attractor towards the vicinity of the saddle. This is
accompanied by a change in the linear stability of the system, and thus the Jacobian. The latter is estimated from the multivariate time series in a sliding window as follows. Consider the underlying dynamical system $\dot{\mathbf{x}}(t) = \mathbf{f}(\mathbf{x}(t))$ with $\mathbf{x} \in \mathbb{R}^d$, and the observed discrete time series $\{\mathbf{x}(1), \mathbf{x}(2), ..., \mathbf{x}(N)\}$, where $N$ is the window size. The linearization of the dynamical system around the point $\mathbf{y}$ is

$$\dot{\tilde{\mathbf{x}}}(t) = \sum_{i=1}^{d} \tilde{x}_i(t) \frac{\partial \mathbf{f}(\mathbf{x})}{\partial x_i}\bigg|_{\mathbf{x}=\mathbf{y}} \tag{A1}$$

with $\tilde{\mathbf{x}}(t) = \mathbf{x}(t) - \mathbf{y}$. Discretized, this can be approximated as:

$$\tilde{\mathbf{x}}(t+1) - \tilde{\mathbf{x}}(t) = \mathbf{x}(t+1) - \mathbf{x}(t) \equiv \Delta\mathbf{x}_t = \delta t \left( \sum_{i=1}^{d} \tilde{x}_i(t) \frac{\partial \mathbf{f}(\mathbf{x})}{\partial x_i}\bigg|_{\mathbf{x}=\mathbf{y}} \right). \tag{A2}$$

In this expression, the factors $\frac{\partial f_j(\mathbf{y})}{\partial x_i}$ are the elements $J_{ji}$ of the Jacobian matrix. They can be estimated with multiple linear regression by sampling different $\Delta\mathbf{x}_t$ as dependent variable and $\tilde{x}_i(t)$ as independent variables for a given $\mathbf{y}$ from within the time series. To this end, we choose $\{\mathbf{x}(t_1), \mathbf{x}(t_2), ..., \mathbf{x}(t_M)\}$ from within the windowed time series, which are the $M$ closest
points to $\mathbf{y}$ in phase space in terms of the distance $D_{y,k} = \sum_{i=1}^{d} [x_i(t_k) - y_i]^2$. For each $\mathbf{x}(t_k)$, we evaluate $\Delta\mathbf{x}_{t_k}$ using the subsequent point in the time series. From the $M$ samples of $\Delta\mathbf{x}_{t_k}$ and $\tilde{x}_i(t_k)$ for $i = 1...d$, we obtain the factors $\frac{\partial f_j(\mathbf{y})}{\partial x_i}$ by multiple linear regression. We then repeat the procedure for every data point in the window as $\mathbf{y}$, and average the results to obtain average Jacobian elements $J_{ji}$ within the sliding window. In this work we chose $M = N/2$. To illustrate how the Jacobian changes in the Stommel model as the system departs the 'off' attractor, we write Eq. 4 in the deterministic case as

$$\frac{dT}{dt} = f(T,S)$$
$$\frac{dS}{dt} = g(T,S). \tag{A3}$$

The corresponding Jacobian of the linearized system is

$$J = \begin{pmatrix} \frac{\partial f(T,S)}{\partial T} & \frac{\partial f(T,S)}{\partial S} \\ \frac{\partial g(T,S)}{\partial T} & \frac{\partial g(T,S)}{\partial T} \end{pmatrix} = \begin{pmatrix} \mathrm{sgn}(T-S) \cdot (S-2T) - 1 & \mathrm{sgn}(T-S) \cdot T \\ \mathrm{sgn}(S-T) \cdot S & \mathrm{sgn}(T-S) \cdot (2S-T) - \eta_3. \end{pmatrix} \tag{A4}$$

Around the attractors, the real parts of both eigenvalues are negative. As the saddle is approached by crossing $q > 0$, the real part of the first eigenvalue becomes positive. Furthermore, the off-diagonal elements of the Jacobian change sign. We propose



this sign change as early warning signal, since it is more robust than the eigenvalues when estimated from noisy data. We define
the early warning signal as

$$\mathcal{J} \equiv \frac{\partial f}{\partial S} - \frac{\partial g}{\partial T}.$$  (A5)

Note that for dynamical systems defined by a gradient of a potential this indicator is not applicable, since it would be 0 in the
whole phase space due to the symmetric Jacobian. Using instead just one of the diagonal elements as indicator still gives good

early warning possibilities with roughly half the statistical power due to the smaller amount of information retained. For time
series from unknown dynamical systems, changes in the individual elements could be monitored simultaneously, potentially
after embedding in case of univariate time series.

*Author contributions.*  All authors contributed to the design of the research and interpretation of the results. J.L. performed the research and
wrote the paper.

*Competing interests.*  The authors declare no competing interest.

*Acknowledgements.*  This is a contribution funded by the Villum Foundation (Grant 17470), the European Union´s Horizon 2020 research
and innovation Programme under the Marie Sklodowska-Curie Grant agreement No 643073, and the European Union's Horizon 2020 project
Tipping Points in the Earth System (Grant 820970 ).



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
