# Peer review of "Abrupt climate change as rate-dependent cascading tipping point"

_Earth System Dynamics, 2021_

## Author Comment (AC1)

We thank the referee for the thorough evaluation of the manuscript and the interesting points raised. These will help improve the clarity of the paper. A point to point response follows, where the referee comments are in *italic* and our respective answers given below.

*The authors identify the importance of sea-ice for the occurrence of abrupt climate changes and couple its dynamics into a Stommel box model. The authors may want to put their model with its inherent dynamic mechanisms in context to other similar attempts highlighting the different implied dynamical mechanisms; for example the conceptual models considered by Boers et al, Proc Natl Acad Sci 115(47):E11005–E11014 and by Gottwald, Clim Dyn (2021) 56:227-243.*

We agree that it is appropriate to add some comments regarding similar efforts with conceptual models. This will be added in the introduction. The work by Gottwald considers sea ice as an intermittent thermal insulator to the polar ocean, represented by the Stommel model, using a chaotic Lorenz '84 model to act as effectively stochastic forcing. The sea ice component then becomes a deterministic approximation of a correlated additive and multiplicative noise process, extremes of which can trigger temporary excursions of the Stommel model. This is similar in spirit to our stochastic forcing of the sea ice component. However, we include a deterministic underlying parameter shift (e.g. changes in ice sheet extent and height) as the main cause of the abrupt transitions, which is in our opinion more in line with existing evidence of the mechanism of DO events. Boers et al also consider a coupling of sea ice/ice shelf to an ocean box model. Here, the sea ice evolves due to a prescribed piecewise-linear feedback, which leads to self-sustained oscillations. Our model differs in that the sea ice dynamics also involve a tipping point, and the ocean component features rate-induced tipping, thus giving the possibility of the cascades of tipping points discussed in the manuscript.

As a result, our study differs from previous studies both in terms of the dynamical mechanism, as well as the interpretation of the underlying driver of the abrupt transitions.

*The authors tune the parameter h in (5) to allow for what they coin smooth bifurcations. Are there observations suggesting that transitions are smooth or non-smooth? Also, it might be helpful to show the two cases of non-smooth and smooth in Figure 3 (and also for the Stommel model). This would clarify what the authors mean by smooth and non-smooth bifurcations; depending on the background of the reader these terms may invoke different associations. Might also be worthwhile defining this in the manuscript.*

We acknowledge that the slightly loose use of the word "smooth" causes some confusion, which we will resolve with more careful wording and some additional explanations in the revised manuscript.

First, we are actually not attempting to coin the term "smooth bifurcation", we just say that in the sea ice model a larger value for h gives a smoother transition of the albedo. This results in a "smoother" appearance of the bifurcation diagram, in the sense of a more "rounded" S-curve of the fold-fold bifurcations. The extreme case for $h \to 0$ would be a Z-curve instead of an S-curve. What cannot be seen from the bifurcation diagram is that in the less smooth case the curvature of the underlying potential only starts to change significantly as one get very close to the bifurcation, limiting the detectability of critical slowing down. We will consider using terminology other than "smooth" here to distinguish from the following.

On the other hand, the Stommel model does have a truly "non-smooth" bifurcation, which comes from a discontinuity in the flow (it is a non-smooth dynamical system).

We will more carefully define our usage of the word non-smooth in this case. The main argument to do so is already given in the paper: The stable and unstable fixed points meet in a cusp, as opposed to a fold as is the case for a "smooth" bifurcation (see Fig. 4).

As a result, the attractor and saddle can come very close to each other already significantly far away from the bifurcation point. In the paper, we consider the shortest distance of the basin boundary to the attractor as a function of the control parameter.

Figure 7b can then be understood to define the smooth versus non-smooth bifurcation. To make this even clearer, we will include another figure as a subfigure (or in the supplemental material), which shows how the fixed points move as eta_1 is changed, and how they merge tangentially in the bifurcation, together with the real part of the first eigenvalue of the Jacobian to show the non-smoothness of the flow. This will help illustrate how the non-smoothness of the flow is related to the collision of the fixed points in a cusp.

Regarding the choice of h and real-world observations, it is difficult to say how smooth the albedo transition should be considered to be. This depends on what is actually modeled by the sea ice variable.

Since we are modeling a large ocean basin, we considered it more appropriate to use a more gradual albedo transition, corresponding to a wider range of partial sea ice cover.

The choice of h does not change any of our results however, besides the fact that for lower values of h (and thus a "less smooth" bifurcation diagram), it would be more difficult to detect a critical slowing down in the sea ice variable, which is however not a main focus of the paper. We will add bifurcation diagrams for smaller versus larger values of h in the Supp material.

*It was not clear to me how their model allows for the succession of abrupt climate changes such as the DO events mentioned in the introduction. The model seems to capture only single transitions. Can the authors comment on this?*

Indeed the model in its present form only captures individual warming transitions. What we tried to point out in the manuscript is that the parameter shift itself can be modeled by a further slow variable, which could be a simple negative feedback reflecting, e.g., the influence of the AMOC on the ice sheets. The ice sheets could then drive the sea ice by their influence on atmospheric circulation. This allows for oscillations in between the 'on' and 'off' states, and thus successive abrupt climate changes. Many other extensions are possible, which could capture a variety of different dynamical mechanisms, such as excitability. We will try to be more explicit about this in the revised manuscript.

*Regarding the new proposed warning indicator J. Am I correct in thinking that the reason why looking at the individual elements of the Jacobian rather than at the eigenvalues of the Jacobian is that the estimation of each element is done via finite-differencing (which is a bad estimator for noisy data) and calculating the eigenvalues exacerbates this via multiplication?*

Indeed this is one of the reasons, as also mentioned on page 17 in the manuscript. The other reason being a bias in the estimation of the elements, which affects the eigenvalues systematically (Fig. S2). This could be due to the method of finite-differencing. We will investigate whether there are better methods in future studies, which will be more method-heavy and thus not suitable to include here. This does however not change the general idea of the early-warning method.

*Figure 1. What are the values of \eta_3 in (b) and of \eta_1 in (c)*

Will add the values to the caption.

*Line 266: andBoers —> and Boers*

Ok.

---

## Author Comment (AC2)

We thank the referee for the evaluation of the manuscript. The suggestions will help significantly improve the readability of the paper. A point to point response follows, where the referee comments are in *italic* and our respective answers given below.

*The paper is well written, but I still find it difficult to read. I encourage the authors to present a clearer "take-home-message."*

There are several take-home messages, which we will try to bring out more concisely and clearly.

1. Recent studies indicate that past abrupt climate change may have arisen as a cascade of tipping points in climate subsystems. We synthesize this into a conceptual model.

2. We suggest that in general rate-induced tipping makes cascading tipping events more likely, as demonstrated in the conceptual model

3. The analysis of the model yields new insights relevant to dynamical systems in general, which relate to a) "soft" tipping points and b) early-warning for rate-induced tipping.

*I have a few suggestions and questions that the authors may use in their revision:*

    *1. Why not present and analyze the coupled model (Eq. 6) before discussing rate-induced tipping in the Stommel model? I feel that a "standard analysis" of the model in Eq. 6 is missing? You have a relatively simple dynamical system and one control parameter (R). Don't you have a simple saddle-node bifurcation in the three-dimensional system?*

Indeed, we decided to leave out a standard bifurcation analysis of the coupled system. Since the coupling is unidirectional, it is much clearer to regard the separated bifurcation diagrams of I with respect to R and (T,S) with respect to eta_1(I).
The bifurcation diagram in the coupled system is a "quadruple" fold. This holds in general when two systems with a double-fold are coupled unidirectionally. See Fig. 1 in Dekker et al. Earth Syst. Dynam., 9, 1243–1260, 2018, for an example of this. Since our model features Heaviside functions, some of these folds are "non-smooth", which makes the bifurcation diagram quite difficult to read.
Further, we do not explore the full quadruple fold here, since only the cascade of sea ice collapse and ocean circulation resurgence is regarded. To summarize, for our purposes it is sufficient and preferable to present the individual bifurcation diagrams of the sub-systems, but we can offer to add a figure with a bifurcation diagram to the Supplemental material.

    *2. As you go into more detail, is it possible to be more precise? I feel that it becomes very descriptive.*

We are unsure what exactly is meant here, but will try to generally improve the clarity and rigor where possible.

> *3. How important is it that the Stommel model has a bistable regime? What would happen if you just coupled the sea-ice model to a simpler model with a smooth transition between "modes".*

It is important, since otherwise the dynamics would be qualitatively different. First, there would not be any rate-dependent phenomena, which is a main focus of the paper. Second, there would be no hysteresis, which is a crucial feature that is typically invoked in the stadial-interstadial dynamics (even though we do not specifically make use of the hysteresis here since we model only warming). Finally, a main aim of the paper is to investigate how cascades of tipping points may have played a role in past (and potentially future) abrupt climate change, and how this may be predicted. This must be done with a model that involves tipping points in multiple components.

> *4. I would like to see a discussion of how your proposed EW indicator would work in a "real-data setting".*

We agree that it is helpful to add some comments regarding this in the revised manuscript. The main steps would be time series embedding, choosing the optimal embedding dimension, and then estimating the Jacobian from the reconstructed multivariate time series.

> *There are a few typos in the manuscript. You'll find them when you read through it carefully. I am looking forward to reading a revised version.*

Ok.